# LTD at amygdalocortical synapses as a novel mechanism for hedonic learning

**Melissa S Haley, Stephen Bruno, Alfredo Fontanini, Arianna Maffei***

Department of Neurobiology and Behavior, SUNY – Stony Brook, Stony Brook, United States

**Abstract** A novel, pleasant taste stimulus becomes aversive if associated with gastric malaise, a form of learning known as conditioned taste aversion (CTA). CTA is common to vertebrates and invertebrates and is an important survival response: eating the wrong food may be deadly. CTA depends on the gustatory portion of the insular cortex (GC) and the basolateral nucleus of the amygdala (BLA) however, its synaptic underpinnings are unknown. Here we report that CTA was associated with decreased expression of immediate early genes in rat GC of both sexes, and with reduced amplitude of BLA-GC synaptic responses, pointing to long-term depression (LTD) as a mechanism for learning. Indeed, association of a novel tastant with induction of LTD at the BLA-GC input in vivo was sufficient to change the hedonic value of a taste stimulus. Our results demonstrate a direct role for amygdalocortical LTD in taste aversion learning.

## Introduction

Sensory stimuli are perceived and processed according to their physicochemical and affective signatures. For example, taste perception depends on the chemical identity of a tastant, as well as on its pleasantness or aversiveness (i.e, its hedonic value). Hedonic value can be modulated by experience. Indeed, the affective value of a palatable gustatory stimulus changes if it becomes associated with gastric malaise, a process known as CTA (*Garcia et al., 1955*). In laboratory settings, CTA can be induced in rats by association of sucrose consumption with a gastric malaise-inducing intraperitoneal (ip) injection of lithium chloride. In naïve rats, this association reliably produces a persistent hedonic shift capable of driving avoidance even after a single conditioning trial. Inactivation studies point to the involvement of the gustatory portion of the insular cortex (*Braun et al., 1972*; *Flynn et al., 1991*; *Yamamoto et al., 1995*; *Cubero et al., 1999*) and the basolateral amygdala (*Morris et al., 1999*) as key regions necessary for CTA. Evidence from these studies is further corroborated by experiments investigating the interaction between these two regions. In naïve animals, analysis of GC neurons' activity following inactivation of BLA indicates that amygdalar inputs to GC provide information about the affective dimension of taste stimuli (*Piette et al., 2012*). Recent studies suggest that the expression of aversive behaviors related to taste depend on the output of GC to the amygdala (*Lavi et al., 2018*; *Schiff et al., 2018*; *Kayyal et al., 2019*).

While the involvement of GC, BLA, and their connections in CTA is well-established, the synaptic changes underlying this form of learning remain unclear. Different experimental approaches produced results that are hard to reconcile. In GC, analysis of molecular markers for synaptic plasticity was brought as evidence that CTA induced long term potentiation (LTP) of GC synapses (*Rosenblum et al., 1993*; *Escobar and Bermúdez-Rattoni, 2000*; *Berman and Dudai, 2001*; *Escobar et al., 2002*; *Gal-Ben-Ari and Rosenblum, 2011*; *Rodríguez-Durán et al., 2011*; *Guzman-Ramos and Bermudez-Rattoni, 2012*). Nevertheless, analysis of GC and BLA neurons' spiking activity in awake rodents reported a decrease in firing rate in both regions following CTA (*Grossman et al., 2008*) – a finding that is inconsistent with the hypothesized induction of LTP.

*For correspondence:
arianna.maffei@stonybrook.edu

**Competing interests:** The authors declare that no competing interests exist.

Here we use a combination of immediate early gene immunohistochemistry, patch clamp electrophysiology, and optogenetic stimulation to study plasticity of amygdalocortical circuits in CTA. We demonstrate that CTA reduces activity in GC, and that this decrease is accompanied by long term depression (LTD) at BLA synapses onto pyramidal neurons in layer 2/3 of GC. Finally, we identify patterns of activity that induce LTD at the BLA-GC input and show that substituting the gastric malaise with this pattern at BLA-GC terminal fields in vivo following sucrose consumption is sufficient to reduce the preference for this tastant. Our results provide the first direct evidence that CTA depends on LTD at amygdalocortical synapses, and that this plasticity alters the affective dimension of a sensory stimulus.

## Results

In order to investigate whether learning induces synaptic plasticity at amygdalocortical synapses in GC, we trained male and female rats in a conditioned taste aversion (CTA) paradigm where novel, palatable sucrose (0.1M in water) was paired with gastric malaise induced by an intraperitoneal (ip) injection of lithium chloride (LiCl, 0.15M, 7.5 mL/kg body weight; *Figure 1A*). Animals were placed on a water restriction schedule where they received 15 min access to water from a drinking spout in an experimental chamber in the morning, followed by one-hour access to water in their home cages 4 hr later. After four days of habituation, on day 5 and 7, the morning water was replaced with 0.1M sucrose, and the animals received an ip injection of LiCl immediately after consumption. Two conditioning sessions were used to ensure learning and consolidation of the aversive memory. In the control group (*Figure 1B*), pseudo-conditioned animals were exposed to both the US and CS, but they received the LiCl injection the night before sucrose access, and therefore did not form an association between the gastric malaise and tastant. Learning was assessed on day 8 using a 2-bottle test. Average fluid consumption across habituation days for each group was comparable, and animals in both groups initially found the sucrose solution to be palatable, consuming more on the first conditioning day compared to baseline water intake (*Figure 1A–B*, *Figure 1—figure supplement 1*). However, this preference shifts in the CTA group following the LiCl injection, as indicated by decreased consumption of sucrose on the second conditioning day, and a strong preference for water at testing (*Figure 1A*). A significant difference in the aversion index (AI), calculated as ((water − sucrose)/total volume) (*Figure 1C*; water preferred for 0 < AI < 1; sucrose preferred if −1 < AI < 0), could be seen in animals that received an ip injection of LiCl following tastant exposure compared to animals that received LiCl dissociated from the tastant exposure. Following the 2-bottle test, brain tissue was harvested for either immunohistological processing or acute slice recording experiments. To ensure that the 2-bottle test did not initiate CTA extinction, in a subset of rats we repeated the 2-bottle test on two consecutive days (*Figure 1—figure supplement 2A*). Results from the 2-bottle tests (*Figure 1—figure supplement 2C*) and aversion indices on test day and 24 hr later were comparable (*Figure 1—figure supplement 2D*). These results indicate that in our paradigm, CTA was reliably induced, and the 2-bottle test did not initiate an extinction process.

### Expression of CTA learning results in decreased recruitment of L2/3 EXC neurons

To begin assessing the effects of CTA on GC, we quantified neurons expressing the activity-dependent immediate early genes (IEGs) c-Fos and EGR1 1 hr after the 2-bottle test (*Figure 1D–F*).We focused our analysis on pyramidal neurons in layers 2/3 of GC, as our previous work showed that they receive a powerful input from BLA (*Haley et al., 2016*). We used c-Fos and EGR1 on the same groups of control and CTA rats to cross validate our results and ensure consistency of effect across animals and conditions. There was a significant decrease in the number of c-Fos and EGR1 positive neurons in the CTA group vs. control (*Figure 1F*), indicating that expression of the learned aversion to sucrose decreased GC neuronal activation. This effect is consistent with previous reports showing a reduction in GC spiking activity (*Grossman et al., 2008*).

### CTA learning does not affect the intrinsic excitability of L2/3 pyramidal neurons in GC but decreases spontaneous synaptic drive

The decrease in the number of IEG-labeled neurons following CTA suggests that the associative learning between taste and malaise induced plastic changes in this region of GC, but does not

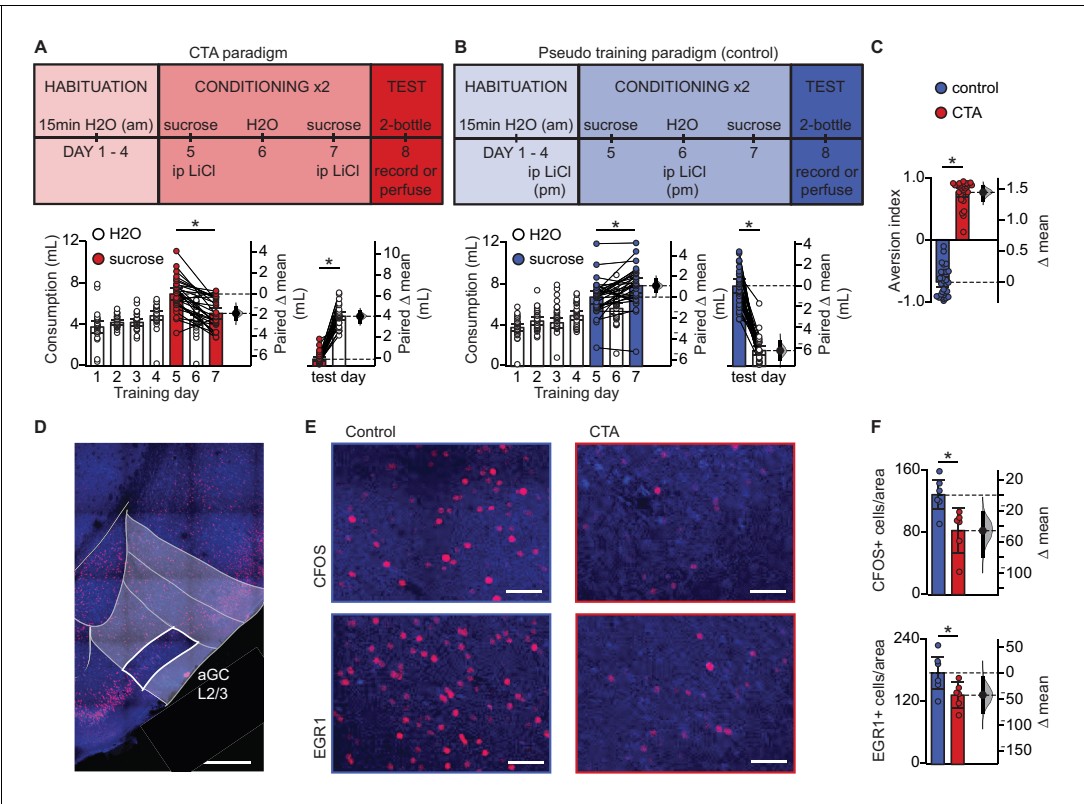

**Figure 1.** Conditioned taste aversion learning differentially recruits EXC neurons in L2/3 aGC. (**A**) Behavioral paradigm schematic. Animals in the control group (N = 30) received ip LiCl (7.5 mL/kg of 0.15M LiCl) the evening prior to exposure to a sucrose solution. Animals in the CTA group (N = 28) received ip LiCl after sucrose exposure. (**B**) Fluid consumption across training days for each group. Rats in both groups showed an initial preference for sucrose over water which shifts after conditioning (Control: C1 sucrose 6.66 ± 0.63 mL vs. C2 sucrose 7.75 ± 0.70 mL, $p<10^{-3}$; CTA: C1 sucrose 6.93 ± 0.61 mL vs. C2 sucrose 5.06 ± 0.53 mL, $p<10^{-7}$) and at testing (2-bottle test, Control: sucrose 7.72 ± 0.64 mL vs. H2O 1.52 ± 0.45 mL, $p<10^{-14}$; CTA: sucrose: 0.66 ± 0.22 mL vs. H2O 4.79 ± 0.38 mL, $p<10^{-16}$). (**C**) Aversion index on test day (Control −0.68 ± 0.08 vs. CTA 0.76 ± 0.07, $p<10^{-32}$). (**D**) Representative image depicting region of interest (ROI, L2/3 aGC, white box). Scale bar = 500 µm. Red: immediate early gene; blue: Nissl counterstain. For details on imaging procedures see Materials and methods. (**E**) Representative images of c-Fos expression (top) and EGR1 expression (bottom) following 2-bottle testing from control group (left) and CTA group (right). Scale bar = 100 µm. (**F**) Average counts of c-Fos expressing and EGR1 expressing neurons normalized to the area of the ROI. (c-Fos [N = 6]: Control 128.13 ± 9.46 positive nuclei/0.1 mm² vs. CTA 82.21 ± 11.73, p<0.01; EGR1 [N = 6], control: 175.90 ± 15.65 vs. CTA: 133.03 ± 10.10, p<0.05). IEG expression was not significantly correlated with sucrose consumption for any group (Control: c-Fos, $r_s$ = 0.60, p=0.21; EGR1, $r_s$ = 0.43, p=0.40; CTA: c-Fos, $r_s$ = −0.37, p=0.47; EGR1, $r_s$ = 0.60, p=0.21). * indicates p≤0.05. Error bars ± 95% CI. The source data reported in the figure are in the *Figure 1—source data 1*.

The online version of this article includes the following source data and figure supplement(s) for figure 1:

**Source data 1.** This file contains source data and statistics presented in each panel of *Figure 1*, *Figure 1—figure supplement 1* and *Figure 1—figure supplement 2*.

**Figure supplement 1.** Baseline water consumption and initial sucrose palatability do not differ between groups.

**Figure supplement 2.** Stability of CTA learning across testing days.

indicate which form of plasticity these changes represent (*Inberg et al., 2013*; *Gandolfi et al., 2017*). Indeed, the effect could result from changes to the intrinsic properties of GC neurons, affecting their overall excitability and input/output function. Alternatively, the decrease in expression could be associated with reduced synaptic efficacy, either at specific inputs, such as BLA, or with altered overall synaptic drive onto these neurons.

We first assessed the effect of CTA on the input/output function of pyramidal neurons in L2/3 of GC. Rats were trained on the control or the CTA paradigm (*Figure 1A–B*) starting at postnatal day 28 (P28). Following the 2-bottle test, acute slice preparations were obtained for whole-cell patch clamp recordings (*Haley et al., 2016*). Excitatory neurons were identified online based on their firing response to steady-state current injection, as well as their pyramidal morphology under DIC optics. Recorded neurons were filled with biocytin for post-hoc confirmation of morphology and location,

and lack of immunostaining for the GABAergic neuronal marker, GAD-67. All recorded neurons included in the analysis were in L2/3, showed pyramidal morphology, and were negative for GAD67 (*Figure 2A*). To test for possible changes in input/output function, we quantified the frequency of action potentials as a function of the amplitude of injected steady-state current steps (*Figure 2B,C*; FI curve, increments of 50 pA). CTA did not affect the input resistance (IR) of GC neurons and left the input/output curve unaltered, indicating a lack of changes in intrinsic properties (*Figure 2C*). These data indicate that following CTA expression, the decrease in the number of GC neurons expressing IEGs is not related to learning-dependent plasticity of intrinsic membrane properties.

Next, we examined if the decreased IEG activation following CTA expression could result from changes in synaptic drive. We performed patch clamp recordings in voltage clamp using a cesium based internal solution that allowed for the isolation of spontaneous excitatory (sEPSCs) currents by holding neurons at the reversal potential for chloride ($-50$ mV in our experimental conditions) (*Figure 2D–E*). Analysis of spontaneous synaptic events unveiled a significant decrease in the cumulative amplitude distribution of spontaneous excitatory synaptic currents, but no change in their frequency (*Figure 2E*). The decrease in amplitude of spontaneous events was more pronounced for larger events and was sufficient to result in a significant reduction in spontaneous excitatory synaptic charge (*Figure 2D*). A change in synaptic drive onto these neurons could potentially mask a change in intrinsic membrane properties, so we repeated the experiments in *Figure 2C* in the presence of ionotropic receptor blockers (20 μm DNQX, 50 μm AP5, 20 μm picrotoxin). No differences in intrinsic membrane properties were unveiled following bath application of the synaptic blockers (*Figure 2—figure supplement 1*). Our results exclude the possibility that CTA altered intrinsic excitability and point to decreased excitatory synaptic transmission as a possible plasticity mechanism for this learning paradigm. Furthermore, given that CTA-induced effects on synaptic drive were detectable in L2/3 pyramidal neurons sampled across the extent of GC, it is highly unlikely for them to depend on selective plasticity on a clustered subset of neurons (*Wang et al., 2018*).

## CTA learning decreases BLA input onto L2/3 pyramidal neurons in aGC

The reduction in the distribution of large events we observed is consistent with the possibility that CTA may affect afferent inputs onto GC neurons, as these typically have larger amplitudes than recurrent events (*Wang et al., 2013*; *Wang et al., 2019*). Previous work demonstrated that CTA induction is impaired by inactivation of either GC or BLA (*Braun et al., 1972*; *Bermudez-Rattoni and McGaugh, 1991*; *Morris et al., 1999*), suggesting that the connection between these regions may be a fundamental player in this form of learning. We therefore asked whether the BLA-GC input onto pyramidal neurons may be affected by CTA. To do that, we used an optogenetic approach to selectively activate BLA terminal fields while recording pyramidal neurons in the superficial layers of GC. As extensively characterized in our previous study (*Haley et al., 2016*), we injected AAV9.CAG.ChR2-Venus.WPRE.SV40 (*Petreanu et al., 2007*) in the BLA (2.1 mm posterior to bregma; 4.7 mm lateral to midline; 7.0 mm below the pia) of P14 rats. CTA training began 14 days after surgery, at P28. The position of the injection site, and expression of the construct in BLA terminal fields within GC were verified histologically (*Figure 3A*). Lack of retrograde transport of the construct was verified by the lack of backfilled somata in GC. Consistency of expression across preparations was assessed as indicated in the Materials and methods (*Figure 3B*), and in our previous work (*Wang et al., 2013*; *Kloc and Maffei, 2014*; *Haley et al., 2016*; *Wang et al., 2019*). Following the 2-bottle test, acute slices were prepared (1.5 mm to bregma) and patch clamp recordings were obtained from visually identified L2/3 pyramidal neurons. Brief (5 ms) pulses of blue light (470 nm) were delivered with an LED mounted on the fluorescence light path of an upright microscope and light-evoked excitatory postsynaptic currents (BLA-EPSCs) were recorded. The monosynaptic nature of BLA-EPSCs was verified as described in our previous work (*Haley et al., 2016*), and an input/output curve for BLA-EPSCs was obtained using light stimuli of increasing intensity (0.2–4.4 mW, measured at the output of the objective). As shown in *Figure 3C*, BLA-EPSC amplitude was reduced following CTA, consistent with the reduction in spontaneous excitatory charge. Recordings in current clamp unveiled that the decreased amplitude of BLA-EPSCs did not reduce the average BLA-evoked EPSP (*Figure 3D*). However, in both groups, there was a strong correlation with the amplitude of BLA-EPSCs and BLA-EPSPs onto individual cells (*Figure 3E*). We observed no differences between the control and CTA groups in the proportion of cells that received input from BLA, or in the paired-pulse ratio (PPR), coefficient of variation (CV), or decay tau of the evoked BLA-

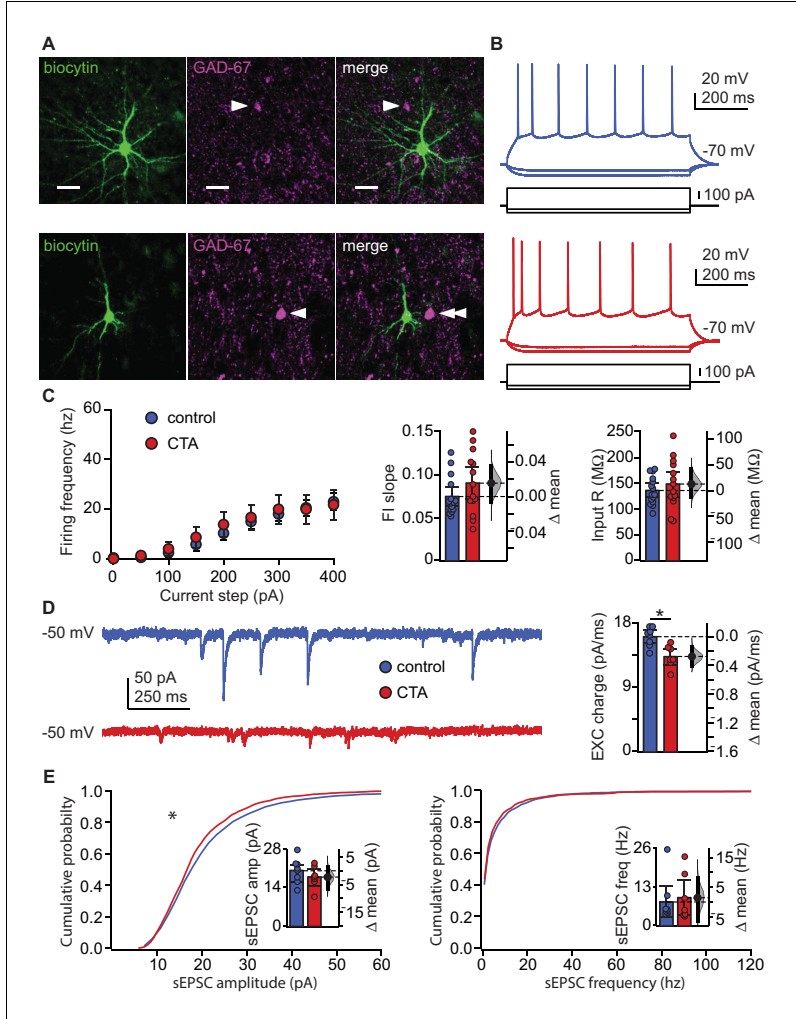

**Figure 2.** CTA learning does not affect intrinsic excitability but reduces excitatory synaptic drive onto EXC neurons in L2/3 aGC. (**A**) Images of example recorded neurons from a control animal (top) and CTA animal (bottom). Left: biocytin (green), middle: GAD-67 (magenta), right: Merge showing lack of co-localization. Scale bar = 25 μm. (**B**) Steady-state firing patterns of neurons from a control animal (top) and CTA animal (bottom). (**C**) FI curves from control (N = 15 rats, 3–4 cells/animal) and CTA animals (N = 14 rats, 3–4 cells/animal). Average FI slope and input resistance did not differ between groups (FI slope: Control 0.07 ± 0.01 vs. CTA 0.09 ± 0.02, p=0.17; Input Resistance: Control 136.47 ± 13.62 MΩ vs. CTA 148.97 ± 24.59 MΩ, p=0.38). (**D**) Sample traces of sEPSCs (holding = −50 mV) onto a control neuron and CTA neuron. Average EXC charge was larger in the control group (Control, [N = 8 rats, 3 cells/animal] 16.09 ± 0.90 pC vs. CTA [N = 7 rats, 3–4 cells/animal] 13.32 ± 1.14 pC, p<0.01). (**E**) Cumulative probability plots showing a decrease in sEPSC amplitude (left), but not sEPSC instantaneous frequency (right) in the CTA group (sEPSC amplitude [n = 24 cells/group, α = 0.0004]: Control vs. CTA, $p<10^{-11}$; sEPSC frequency [n = 24 cells/group, α = 0.0004]: Control vs. CTA, p=0.02). Average sEPSC amplitude and frequency (insets) did not differ between groups (sEPSC amplitude: Control, [N = 8 rats, 3 cells/animal] 20.23 ± 3.12 pA vs. CTA [N = 7 rats, 3–4 cells/animal] 17.94 ± 2.96 pA, p=0.32; sEPSC frequency: Control, [N = 8 rats, 3 cells/animal] 8.14 ± 5.17 Hz vs. CTA [N = 7 rats, 3–4 cells/animal] 9.55 ± 5.90 Hz, p=0.75). * indicates p≤0.05. Error bars ± 95% CI. The source data reported in the figure are in the *Figure 2—source data 1*.

The online version of this article includes the following source data and figure supplement(s) for figure 2:

**Source data 1.** This file contains source data and statistics presented in each panel of *Figure 2*, *Figure 2—figure supplement 1*.

**Figure supplement 1.** Changes in synaptic drive do not mask changes in intrinsic excitability.

EPSC (*Figure 3—figure supplement 1*). Since BLA provides a powerful glutamatergic input to L2/3 neurons (*Haley et al., 2016*), the decrease in BLA-EPSC amplitude strongly suggests that GC neurons are activated less effectively by incoming inputs. This effect may explain the reduced expression of IEGs in *Figure 1D* and is consistent with reports of decreased GC neurons' activity in vivo (*Grossman et al., 2008*).

## Induction of LTD at BLA-GC synapses is occluded following CTA learning

We hypothesized that the decrease in BLA-EPSC amplitude may be evidence for induction of LTD at BLA-GC synapses onto L2/3 neurons. We first probed the capacity for plasticity of these synapses by identifying a pattern of activity that could induce LTD at this input in control animals. We designed induction paradigms that allowed for stimulation of BLA terminal fields with a phasic or a tonic pattern of activity. Both patterns of BLA neuron activity have been recorded in vivo in awake rodents (*Fontanini et al., 2009*; *Parsana et al., 2012*) and are thought to be associated with different behavioral states (*Parsana et al., 2012*). The effect of such activity patterns on spontaneous GC activity was reported in our previous study (*Haley et al., 2016*). Patch clamp recordings were obtained from L2/3 neurons in GC of slices obtained from control and CTA rats, prepared right after the 2-bottle test. Following acquisition of a 10 min baseline recorded in voltage clamp at −70 mV, we paired presynaptic 20 Hz bursts of light pulses to activate BLA afferents with depolarization of the postsynaptic GC neuron in current clamp, then continued to record neurons in voltage clamp at −70 mV for at least 40 min post-induction. This phasic induction pattern reliably induced LTD at BLA-GC synapses in 56.52% of the recordings from control rats (*Figure 4B*). The remaining 43.48% of recordings showed either no change (21.74%) or potentiation (21.74%), indicating that BLA-GC synapses are capable of expressing both synaptic depression and potentiation. LTD induced by phasic BLA terminal field activation did not show changes in PPR nor CV (*Figure 4C*), indicative of a postsynaptic site

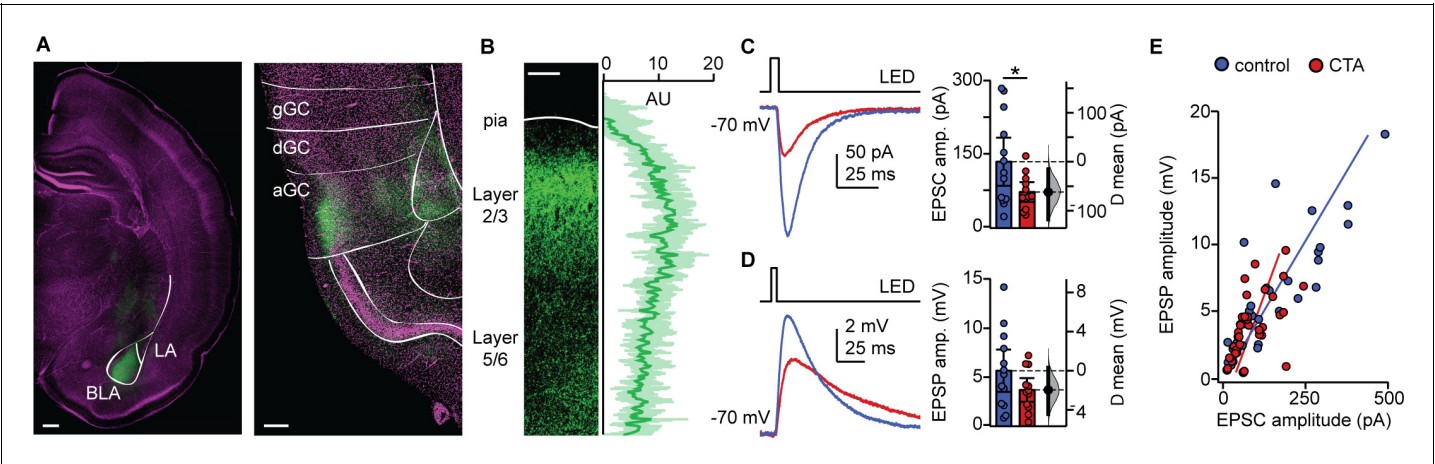

**Figure 3.** CTA decreases BLA input onto L2/3 pyramidal neurons in aGC. (**A**) Example of histological verification of injection site and expression of opsin construct in BLA terminal fields in aGC. Left: Image of ChR2-Venus expression at BLA injection site (scale bar = 500 μm; BLA, basolateral nucleus; LA, lateral nucleus); right: image of ChR2-Venus expression of BLA afferent fibers in aGC (scale bar = 250 μm; green: ChR2-Venus, magenta: Hoechst; gGC, granular GC; dGC, dysgranular GC; aGC, agranular GC). (**B**) Left: Image of ChR2-VENUS expression in BLA afferent fiber in aGC. Scale bar = 250 μm. Right: Calibration curve of ChR2-VENUS intensity across lamina in aGC (±1 SD). (**C**) Sample traces of optogenetically-evoked BLA-EPSCs. BLA-EPSC amplitudes onto CTA neurons were reduced (Control [N = 13 rats, 2–3 cells/animal] 135.42 ± 49.17 pA vs. CTA [N = 12 rats, 2–3 cells/animal] 73.36 ± 20.17 pA, p<0.03). (**D**) Sample traces of optogenetically-evoked BLA-EPSPs. Summary of BLA-EPSP amplitudes onto control neurons and CTA neurons (Control [N = 13 rats, 2–3 cells/animal] 5.67 ± 2.16 mV vs. CTA [N = 12 rats, 2–3 cells/animal] 3.70 ± 1,20 mV, p=0.14). (**E**) Input-output plot of BLA-EPSCs and BLA-EPSPs for the control group and CTA group (Control [n = 30 cells] $r_s$ = 0.84, p<$10^{-6}$; CTA [n = 34 cells] $r_s$ = 0.69, p<$10^{-5}$). * indicates p≤0.05. Error bars ± 95% CI. The source data reported in the figure are in the *Figure 3—source data 1*.

The online version of this article includes the following source data and figure supplement(s) for figure 3:

**Source data 1.** This file contains source data and statistics presented in each panel of *Figure 3*, *Figure 3—figure supplement 1*.
**Figure supplement 1.** CTA does not affect connectivity, PPR, CV, or decay tau.

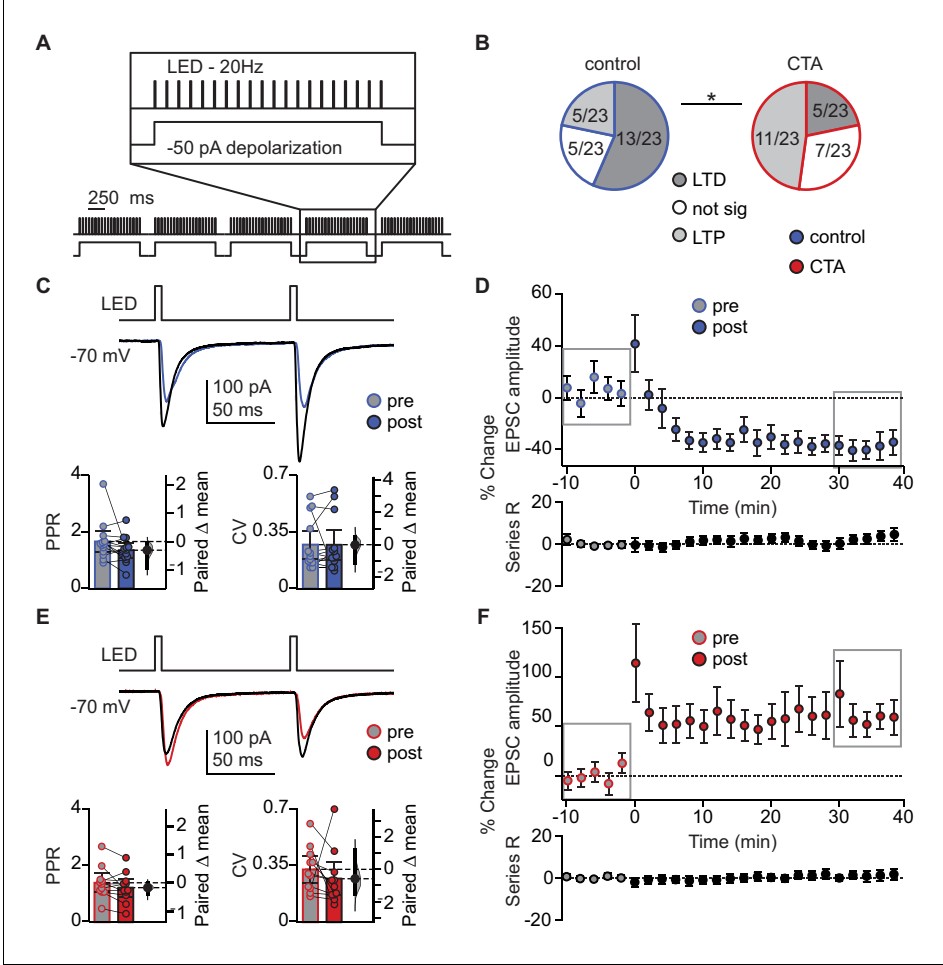

**Figure 4.** CTA learning occludes LTD induction onto L2/3 pyramidal neurons in aGC. (**A**) Schematic of 20 Hz plasticity induction paradigm. BLA terminal fields were activated with 20 bursts of 20 light pulses (5 ms) at 20 Hz delivered every 250 ms, while the postsynaptic neuron was depolarized subthreshold by injection of a 50 pA current step. (**B**) Distribution of the outcome of plasticity induction across all neurons recorded (Control LTD [n = 13 cells from 8 rats] 56.52%, no change [n = 5 cells from 5 rats] 21.74%, LTP [n = 5 cells from two rats] 21.74% vs. CTA LTD [n = 5 cells from five rats] 21.74%, no change [n = 7 cells from 4 rats] 30.43%, LTP [n = 11 cells from 6 rats] 47.83%, $3 \times 2 \chi^2$, p<0.05). (**C**) Sample traces of control group BLA-EPSC before and after 20 Hz plasticity induction. 20 Hz LTD induction did not affect control group PPR or CV (Control [n = 13 cells from 8 rats]: PPR pre 1.66 ± 0.38 vs. post 1.36 ± 0.26, p=0.10; CV pre 0.27 ± 0.09 vs. post 0.27 ± 0.09, p=0.96). (**D**) Time course of change in BLA-EPSC amplitude (Control: % change from baseline −35.66 ± 2.65) and series resistance (Control: % change from baseline 2.57 ± 0.78) following 20 Hz LTD induction in control group. Boxes indicate pre- and post-induction epochs. (**E**) Sample traces of CTA group BLA-EPSC before and after 20 Hz plasticity induction. 20 Hz LTP induction did not affect CTA group PPR or CV (CTA [n = 11 cells from 6 rats]: PPR pre 1.39 ± 0.33 vs. post 1.22 ± 0.31, p=0.09; CV pre 0.33 ± 0.08 vs. post 0.27 ± 0.10, p=0.41). (**F**) Time course of change in BLA-EPSC amplitude (CTA: % change from baseline 62.37 ± 8.55) and series resistance (CTA: % change from baseline 0.94 ± 0.47) following 20 Hz LTP induction in CTA group. Boxes indicate pre- and post-induction epochs. * indicates p≤0.05. Error bars ± 95% CI. The source data reported in the figure are in the *Figure 4—source data 1*.

The online version of this article includes the following source data and figure supplement(s) for figure 4:

**Source data 1.** This file contains source data and statistics presented in each panel of *Figure 4*, *Figure 4—figure supplement 1*.

**Figure supplement 1.** Activity during 20Hz induction protocol and relationship between baseline PPR, CV, or decay tau and post-induction plasticity.

of expression (*Malinow and Malenka, 2002*; *Kessels and Malinow, 2009*), and consistent with the decrease in amplitude, but not frequency, of sEPSCs shown in *Figure 2E*.

We used this induction paradigm to perform an occlusion experiment, testing the possibility that if CTA had saturated LTD at BLA-GC synapses, further LTD induction would result in no additional depression (*Rioult-Pedotti, 2000*; *Maffei et al., 2006*; *Crozier et al., 2007*). LTD induction by phasic patterns of BLA terminal field activity in GC was indeed occluded by CTA (*Figure 4E–F*): the proportion of recordings showing LTD in response to phasic BLA activity was reduced to 21.74% (from 56.52% in control, *Figure 4B*), while that of recordings showing LTP increased to 47.83% (from 21.74% in control, *Figure 4B*). The proportion of recordings in which induction produced no change in synaptic efficacy was not different from control (30.43%, *Figure 4B*).

LTP induced with the phasic plasticity protocol in the CTA group had no effect on either PPR or CV, consistent with a postsynaptic site of plasticity expression (*Figure 4E*). A similar number of cells in the control and CTA groups fired action potentials during the induction protocol, however neither spiking activity during induction, nor the initial PPR, CV, or decay tau, were predictive of the sign of plasticity induced in either group (*Figure 4—figure supplement 1B,C*). The results of the occlusion experiment strongly suggest that LTD at the BLA-GC synapse is associated with CTA learning.

To assess the specificity of the occlusion, we tested the effect of CTA on LTD induction with a different paradigm designed to mimic tonic BLA activity (*Fontanini et al., 2009*; *Parsana et al., 2012*). Here, we paired presynaptic ramping light stimuli to tonically activate BLA afferents (*Haley et al., 2016*), with depolarization of the postsynaptic GC neuron. LTD induction with the tonic BLA activity regime was not occluded by CTA, resulting in LTD in 100% of the recordings in both control and CTA rats (*Figure 5A–F*). This form of LTD was accompanied by a significant reduction in PPR in the control group (*Figure 5C*), and a significant change in both PPR and CV in the CTA group (*Figure 5E*).

This decrease in PPR is consistent with a presynaptic site of plasticity expression, indicating that tonic BLA activity engages a distinct set of plasticity mechanisms that are not consistent with the changes in sEPSC amplitude we observed following CTA. Similar to the 20 Hz protocol, neither spiking during induction, nor initial PPR, CV, or decay tau, were correlated with the degree of plasticity (*Figure 5—figure supplement 1B,C*). These results strongly suggest that successful CTA learning is associated with postsynaptically expressed LTD induced by phasic activation of BLA afferents in GC.

## Phasic activation of BLA terminal fields in GC in vivo eliminated sucrose preference

In view of the selective occlusion of LTD induced by phasic BLA terminal field activation, we asked whether association of novel sucrose and phasic activation of BLA terminal fields in GC would be sufficient to change the hedonic value of the tastant. To do that, we injected the AAV9.CAG.ChR2-Venus.WPRE.SV40 construct (*Petreanu et al., 2007*) in BLA and implanted an optic fiber (400 µm) in GC (paired 20 Hz opto; *Figure 6A*). As control, a group of animals was injected with an AAV9.hSyn.eGFP.WPRE.bGH construct (University of Pennsylvania Vector Core) and implanted with the optic fiber in GC (GFP control). These two rat groups were trained on a modified CTA paradigm in which the LiCl-induced malaise was substituted with phasic activation of BLA terminal fields (*Figure 6C,D*). A third group of rats was injected with ChR2 and received non-contingent phasic activation of BLA terminal fields the evening before sucrose exposure (non-paired 20 Hz opto, *Figure 6E*). Finally, a fourth group of rats was injected with ChR2 and the LiCl-induced malaise was replaced by tonic activation of BLA terminal fields with the ramping light stimulus (*Figure 6F*). All groups showed a comparable initial preference for sucrose over water, including the non-paired 20 Hz opto group, indicating that 20 Hz stimulation alone did not affect sucrose detection or palatability (*Figure 6—figure supplement 1*). However, as shown in *Figure 6D*, only the rat group that received phasic light activation of BLA terminal fields in GC paired with sucrose showed a reduced preference for sucrose. This effect is further supported by the change in AI (*Figure 6G*). Thus, while both the phasic 20 Hz and the ramp tonic paradigm induced LTD in acute slice preparations, CTA was selectively induced by pairing the phasic 20 Hz pattern of activation of BLA afferents in GC with sucrose consumption.

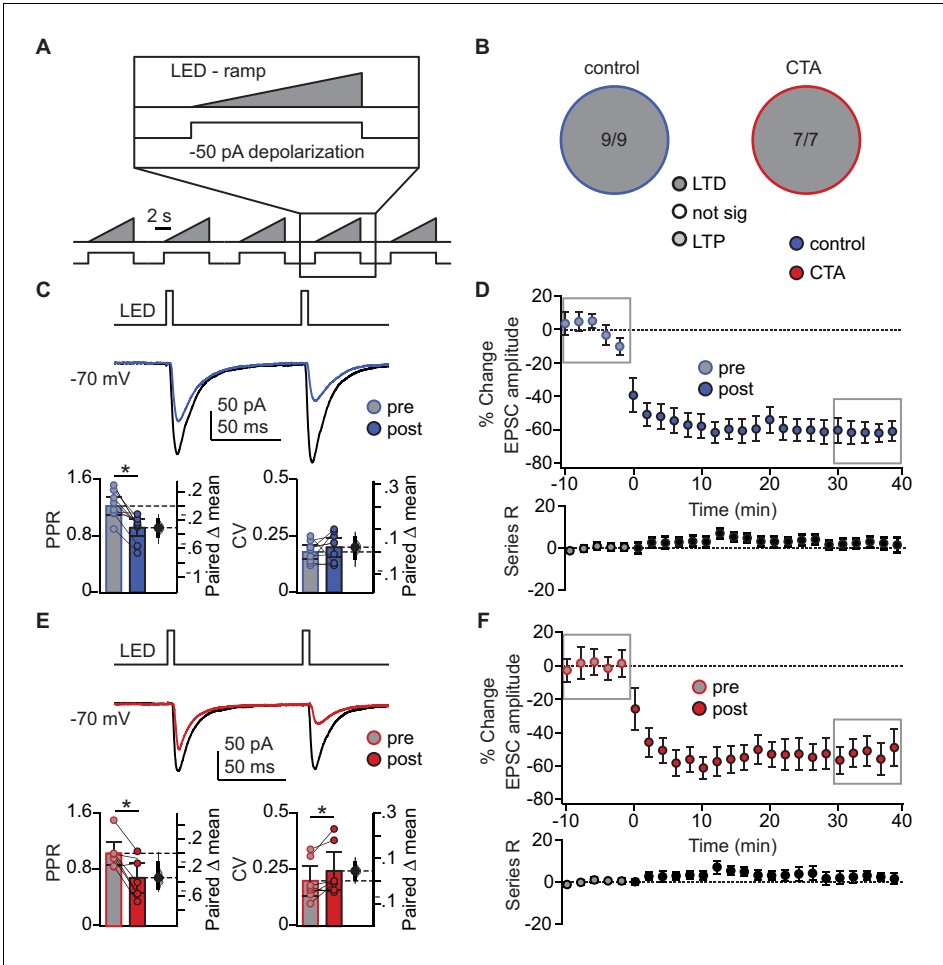

**Figure 5.** Occlusion of LTD following CTA is specific to BLA activity regime. (**A**) Schematic of tonic induction paradigm. BLA terminal fields were activated with 6 s ramping stimuli delivered every 250 ms, while the postsynaptic neuron was depolarized subthreshold by injection of a 50 pA current step. (**B**) Distribution of the outcome of plasticity induction across all neurons recorded (Control LTD [n = 9 cells from 4 rats] 100.00%, vs. CTA LTD [n = 7 cells from 4 rats] 100.00%, $3 \times 2 \chi^2$, p=1.00). (**C**) Sample traces of control group BLA-EPSC before and after ramp plasticity induction. Ramp LTD induction reduced control group PPR but did not affect CV (Control [n = 9 cells from 4 rats]: PPR pre 1.22 ± 0.12 vs. post 0.92 ± 0.12, $p<10^{-3}$; CV pre 0.18 ± 0.03 vs. post 0.20 ± 0.04, p=0.33). (**D**) Time course of change in BLA-EPSC amplitude (Control: % change from baseline −61.87 ± 10.92) and series resistance (Control: % change from baseline 4.37 ± 4.01) following ramp LTD induction in control group. Boxes indicate pre- and post-induction epochs. (**E**) Sample traces of CTA group BLA-EPSC before and after ramp plasticity induction. Ramp LTD induction reduced CTA group PPR and increased CV (CTA [n = 7 cells from 4 rats]: PPR pre 1.03 ± 0.16 vs. post 0.68 ± 0.21, p<0.03; CV pre 0.20 ± 0.07 vs. post 0.24 ± 0.08, p<0.03). (**F**) Time course of change in BLA-EPSC amplitude (CTA: % change from baseline −52.70 ± 6.48) and series resistance (CTA: % change from baseline 2.16 ± 4.79) following ramp LTD induction in CTA group. Boxes indicate pre- and post-induction epochs. * indicates p≤0.05. Error bars ± 95% CI. The source data reported in the figure are in *Figure 5—source data 1*.

The online version of this article includes the following source data and figure supplement(s) for figure 5:

**Source data 1.** This file contains source data and statistics presented in each panel of *Figure 5*, *Figure 5—figure supplement 1*.

**Figure supplement 1.** Activity during ramp induction protocol and relationship between baseline PPR, CV, or decay tau and post-induction plasticity.

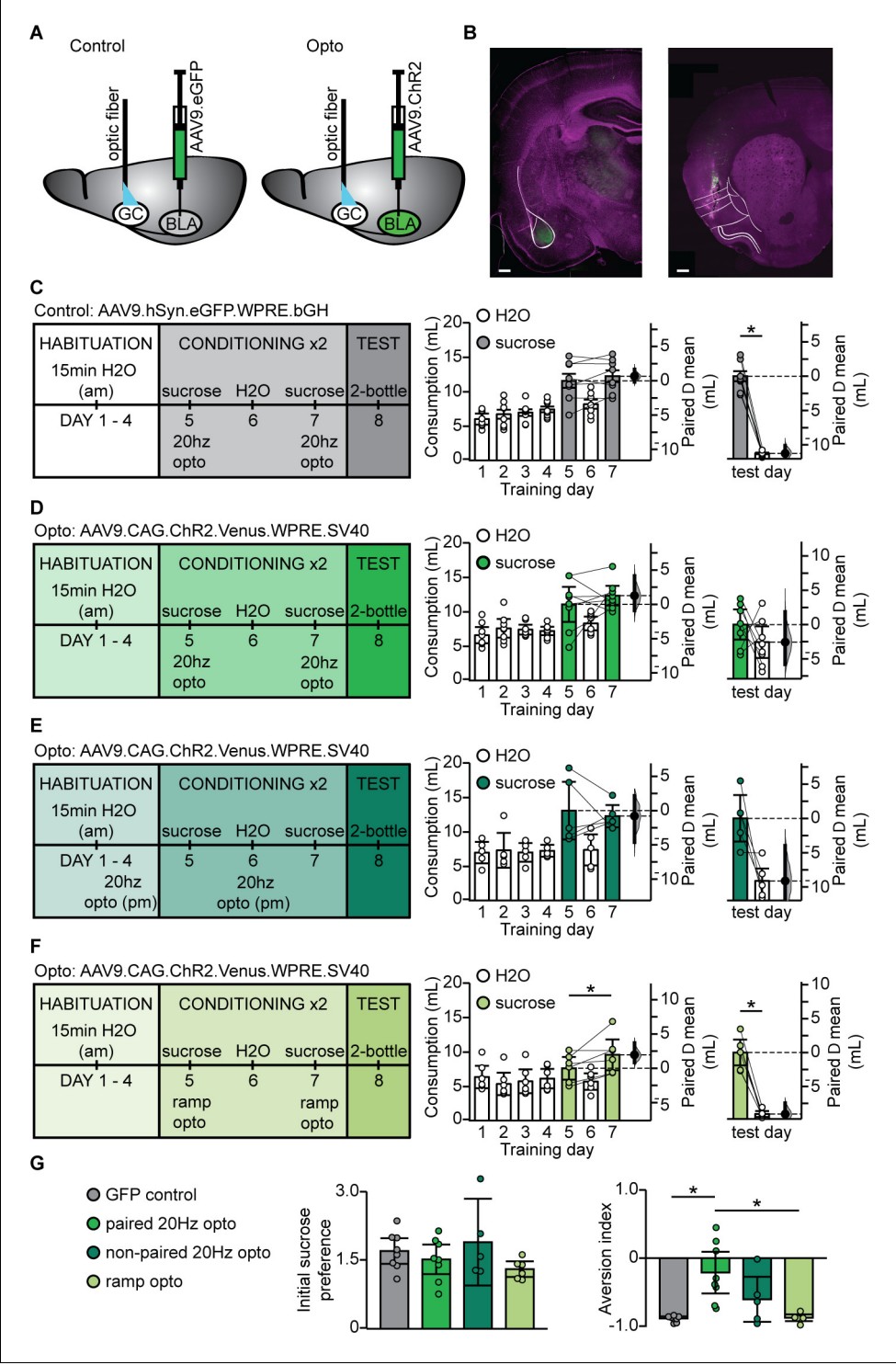

**Figure 6.** BLA terminal field activation in vivo with the phasic 20 Hz pattern eliminated sucrose preference. (**A**) Cartoon of experimental approach. AAV9.hSyn.eGFP.WPRE.bGH: (GFP control group) or AAV9.CAG.ChR2-Venus. WPRE.SV40 (Paired 20 Hz group) were injected in BLA and an optic fiber (400 μm diameter, coated with DiL) was implanted in aGC. (**B**) Correct positioning of injection site in BLA and of optic fiber in aGC were verified histologically. Left, green: injection site of ChR2-Venus; magenta: Hoechst counterstain; Right, green DiL indicating optic fiber tract; magenta: counterstain. White lines delineate anatomical landmarks indicating the location of BLA (left) and GC (right) based on the rat brain atlas (*Paxinos and Watson, 1998*). Scale bar = 500 μm. (**C**) To control for non-specific effects of light stimulation, one group of rats was injected with an AAV9 construct containing only

*Figure 6 continued on next page*

*Figure 6 continued*

the fluorescent tag Venus (GFP control, N = 8 rats). Diagram of behavioral paradigm for CTA in which the LiCl injection was substituted with 20 Hz optogenetic stimulation of BLA terminal fields in GC following sucrose exposure. Fluid consumption across training days for GFP control group. Rats showed an initial preference for sucrose over water which persisted after conditioning (GFP control: C1 sucrose 11.60 ± 1.01 mL vs. C2 sucrose 12.27 ± 0.86 mL, p=0.23) and at testing (GFP control: test day sucrose 12.03 ± 0.73 mL vs. test day $H_2O$ 0.77 ± 0.12 mL, $p<10^{-7}$). (D) Diagram of behavioral paradigm for CTA in which the LiCl injection was substituted with 20 Hz optogenetic stimulation of BLA terminal fields in GC following sucrose exposure for rats that were injected with ChR2 (paired 20 Hz opto, N = 8 rats). Fluid consumption across training days for paired opto group. Rats showed an initial preference for sucrose over water which shifted after conditioning (paired 20 Hz opto: C1 sucrose 11.10 ± 2.55 mL vs. C2 sucrose 12.36 ± 1.43 mL, p=0.41) at testing (paired 20 Hz opto: test day sucrose 7.91 ± 2.10 mL vs. test day $H_2O$ 5.35 ± 2.31 mL, p=0.24). (E) Diagram of behavioral paradigm for CTA in which rats received non-contingent 20 Hz optogenetic stimulation of BLA terminal fields in GC the evening prior to sucrose exposure (non-paired 20 Hz opto, [N = 5 rats]). Fluid consumption across training days for GFP control group. Rats showed an initial preference for sucrose over water which persisted after conditioning (non-paired 20 Hz opto: C1 sucrose 13.11 ± 4.22 mL vs. C2 sucrose 12.31 ± 1.62 mL, p=0.68) and at testing (non-paired 20 Hz opto: test day sucrose 11.98 ± 3.40 mL vs. test day $H_2O$ 2.86 ± 2.33 mL, p=0.07). (F) Diagram of behavioral paradigm for CTA in which the LiCl injection was substituted with optogenetic ramp stimulation of BLA terminal fields in GC following sucrose exposure for rats that were injected with ChR2 (ramp opto, [N = 6 rats]). Fluid consumption across training days for ramp opto group. Rats showed an initial preference for sucrose over water which persisted after conditioning (ramp opto: C1 sucrose 7.64 ± 1.66 mL vs. C2 sucrose 9.60 ± 2.24 mL, p=0.06) and at testing (ramp opto: test day sucrose 9.81 ± 1.89 mL vs. test day $H_2O$ 0.73 ± 0.43 mL, $p<10^{-4}$). (G) Sucrose preference scores (C1 sucrose/average H2O) did not differ between groups (GFP Control 1.70 ± 0.28, paired 20 Hz opto 1.52 ± 0.32, non-paired 20 Hz opto 1.90 ± 0.74, ramp opto 1.30 ± 0.17, 1-way ANOVA p=0.25). Aversion index of the paired 20 Hz opto group differed from both the GFP control and ramp opto on test day (GFP Control 1.70 ± 0.28, paired 20 Hz opto 1.52 ± 0.32, non-paired 20 Hz opto 1.90 ± 0.74, ramp opto 1.30 ± 0.17, 1-way ANOVA $p<10^{-3}$, GFP control vs. paired 20 Hz opto, p=0.001; paired 20 Hz opto vs. ramp opto, p=0.0017). * indicates p≤0.05. Error bars ± 95% CI. The source data reported in the figure are in the *Figure 6—source data 1*.

The online version of this article includes the following source data and figure supplement(s) for figure 6:

**Source data 1.** This file contains source data and statistics presented in each panel of *Figure 6*, *Figure 6—figure supplement 1*.

**Figure supplement 1.** Baseline water consumption and initial sucrose palatability do not differ between groups.

## Discussion

Our data demonstrate that CTA learning leads to reduced activation of the taste responsive portion of the insular cortex and to an LTD-related reduction in strength of the BLA-GC connection. The results also demonstrate that induction of LTD at amygdalocortical synapses onto L2/3 pyramidal neurons with a phasic 20 Hz pattern of stimulation of BLA terminal fields in GC is sufficient for changing the hedonic value of a taste stimulus. These findings confirm the central role of the projection from BLA to GC in CTA, and also reveal that circuit changes following CTA learning do not depend, as previously thought, on strengthening of synaptic connections, but rely on a significant decrease in synaptic input. Previous work reported a substantial change in the cortical representation of the conditioned tastant in GC following aversive learning (*Yasoshima and Yamamoto, 1998*; *Accolla et al., 2007*; *Grossman et al., 2008*). Our work provides novel evidence regarding the circuit mechanisms underlying this reorganization process and identifies long term plastic changes associated with CTA.

### CTA and immediate early gene expression

The behavioral paradigm we used consisted of two rounds of conditioning with taste-malaise pairing to ensure reliable expression of aversion learning. As a readout of the aversive memory, we used a 2-bottle test to determine volume consumption of the conditioned stimulus following learning, and to compute the aversion index between control and CTA animals. The 2-bottle test did not initiate an extinction process, as we show that the aversion index remained stable when animals were tested on two consecutive days. Quantification of IEGs, as a proxy for GC neurons' activity, showed reduced expression of both c-Fos and EGR1 following testing in CTA animals compared to control.

The decrease in IEG expression provides additional support as to the lack of extinction initiation, as extinction has been shown to increase IEG expression in GC (*Hadamitzky et al., 2015*).

In our paradigm, all animals, including controls, are exposed to sucrose twice. While for CTA animals sucrose is always associated with a gastric malaise, in control animals sucrose is a safe, palatable stimulus and animals consume it more than water. In control animals, the consumption of sucrose on the second exposure is higher than the first, consistent with an attenuation of neophobia (*Lin and Reilly, 2012*), an effect that is associated with a reduction in c-Fos expression in GC (*Lin et al., 2012*). In the CTA group, the consumption of sucrose at the second exposure was significantly lower than at the first. The differences in IEG expression we report may reflect a blunted GC activation to sucrose in CTA animals compared to control. Consistent with this possibility, previous studies showed increased IEG expression in the insular cortex following consumption of a novel tastant alone (saccharine), but not in response to a LiCl injection alone (*Wilkins and Bernstein, 2006*; *Bernstein and Koh, 2007*). Thus, our data support the interpretation that the differences in IEG expression are due to animals in the CTA group expressing an aversion to sucrose, rather than the control group learning that sucrose is 'safe'. As IEG expression is considered a proxy for activity, our results are consistent with previously published work showing reduced spiking activity of GC neurons in the palatability epoch following CTA (*Grossman et al., 2008*; *Piette et al., 2012*). Thus, the aversive memory induced by our behavioral paradigm was reliable, allowing us to assess the effects of CTA on the BLA-GC circuit.

## CTA reduced excitatory synaptic drive onto GC neurons

Our analysis of synaptic and intrinsic properties of GC neurons allowed us to parse apart which of these components was responsible for the reduction in GC activity seen with CTA. We show that excitatory spontaneous synaptic charge onto GC pyramidal neurons is reduced following CTA, an effect that points to widespread modulation of synaptic transmission in GC, indicating that CTA affects the circuit in GC broadly. CTA altered synaptic drive and capacity for plasticity at BLA-GC synapses, effects consistent with changes in the activation of GC. These results strongly suggest that decreased synaptic drive onto GC neurons underlies the CTA-dependent circuit reorganization reported in previous work (*Accolla and Carleton, 2008*) and explains the reduced firing rates of GC neurons following CTA (*Grossman et al., 2008*).

Analysis of spontaneous excitatory synaptic events revealed a significant decrease in amplitude, and no changes in frequency. These effects are consistent with a postsynaptic site of expression of CTA-dependent modifications of synaptic efficacy (*Malinow and Malenka, 2002*; *Rumpel et al., 2005*). Further confirmation of the postsynaptic site of expression comes from analysis of BLA-evoked responses onto GC pyramidal neurons. Following CTA, we observed a significant decrease in the amplitude of BLA-evoked responses, but no change in their PPR. Furthermore, the lack of changes in intrinsic excitability indicates that the profound effects taste aversion learning has on GC pyramidal neurons is via modulations at specific synaptic inputs and not an alteration in intrinsic membrane properties. Together, this set of findings confirms the central role of the BLA-GC connection in CTA learning, and demonstrates a role for plasticity of BLA-evoked synaptic responses onto L2/3 pyramidal neurons in hedonic learning.

## Postsynaptically expressed LTD at amygdalocortical synapses as a mechanism for CTA

When we compared the effect of CTA on two different forms of LTD, induced by patterns of stimulation designed to mimic BLA neurons' activity in different behavioral states (*Fontanini et al., 2009*; *Parsana et al., 2012*), we found a selective occlusion of plasticity induced by a phasic pattern of activity. Phasic stimulation induced a postsynaptically expressed form of LTD in control rats, but LTP in the CTA group, suggesting that phasic LTD and CTA share common mechanisms. The occlusion was specific to the phasic induction pattern, as LTD induced with a tonic pattern of BLA terminal field activation, a form of LTD with a presynaptic site of expression, was not affected by CTA. These results point to postsynaptic LTD at BLA-GC synapses as a mechanism for taste aversion learning.

The underlying mechanisms of postsynaptically expressed LTP and LTD have been studied extensively. Signaling cascades leading to the regulation of AMPA receptors in the postsynaptic membrane are thought to mediate changes in synaptic efficacy (*Malinow and Malenka, 2002*;

*Kessels and Malinow, 2009*). Current thinking associates learning with an increase in synaptic efficacy (*Bliss and Collingridge, 1993*; *Rogan et al., 1997*; *Dityatev and Bolshakov, 2005*), and changes in molecular markers for plasticity following CTA have been interpreted as indication of LTP expression (*Shema et al., 2007*; *Gal-Ben-Ari and Rosenblum, 2011*). However, many of the molecular mechanisms involved in CTA (*Berman and Dudai, 2001*; *Gal-Ben-Ari and Rosenblum, 2011*; *Guzman-Ramos and Bermudez-Rattoni, 2012*) can be engaged by both LTP and LTD (*Yilmaz-Rastoder et al., 2011*; *Gandolfi et al., 2017*; *Li and Pozzo-Miller, 2019*). Additional evidence in support of the interpretation that CTA should potentiate BLA input to GC came from studies based on BLA activation with drug infusions or tetanic electrical stimulation during CTA learning. Local field potential (LFP) recordings in GC of anesthetized rodents showed that the induction of LTP of the population signal evoked by tetanic stimulation of the amygdala was occluded by CTA (*Escobar and Bermúdez-Rattoni, 2000*; *Escobar et al., 2002*), suggesting that plasticity at the BLA input to GC may play a role in this learning paradigm. However, this experimental design does not assess whether the baseline response has been affected by CTA, nor which component of the population signal has been modified by CTA. Furthermore, these data do not indicate whether LTP at BLA-GC synapses may be sufficient for learning.

A key feature of our study that distinguishes it from previous work is the use of optogenetic techniques to selectively activate BLA terminal fields within GC. This approach prevented confounds due to recruitment BLA collaterals to other areas which inevitably occurs with somatic activation of BLA neurons (*Klavir et al., 2017*). Furthermore, our study is the first to examine baseline synaptic transmission, and therefore may yield different results from field potential recordings, which lack synaptic resolution, and are difficult to interpret as the circuit components that contribute to the LFP trace in GC have not been described (*Escobar and Bermúdez-Rattoni, 2000*; *Escobar et al., 2002*; *Rodríguez-Durán et al., 2011*). No analysis of changes in synaptic efficacy has been reported up to the present work. In view of our results demonstrating a reduction in the efficacy of BLA glutamatergic synapses onto GC neurons following CTA, it is likely that in L2/3 pyramidal neurons, signaling mechanisms for LTD are recruited during CTA.

## LTD at amygdalocortical synapses in GC is sufficient for the induction of CTA

Association of sucrose and phasic stimulation of BLA terminal fields in GC in vivo was sufficient to induce a weak CTA, unveiled as a loss of preferences for sucrose over water during the 2-bottle test. In these experiments, the phasic pattern of stimulation was delivered after sucrose in place of LiCl, thus selectively activating the BLA-GC input in the absence of a gastric malaise. Our results indicate that the pattern of activity for phasic 20 Hz LTD at BLA-GC synapses was sufficient to change the hedonic value of sucrose. The effect was specific to the pairing of sucrose and phasic 20 Hz stimulation of BLA terminal fields in GC, as neither non-contingent stimulation, nor the tonic ramping stimulus affected sucrose preference. While phasic 20 Hz activation of BLA-GC terminal fields in vivo was sufficient for changing the valence of sucrose, it did not induce as strong an aversion as LiCl. This effect was expected, as LiCl induces a gastric malaise that is not recruited in animals that received light stimulation through the optic fiber during CTA training. GC receives extensive visceral input (*Cechetto and Saper, 1987*) which is not activated during BLA terminal field stimulation. Furthermore, previous studies have shown no change in IEG expression in the insular cortex following LiCl-induced illness, suggesting that processing of information about the US primarily occurs in other parts of the central gustatory system (*Wilkins and Bernstein, 2006*; *Bernstein and Koh, 2007*). Our findings therefore suggest that the plasticity of BLA-GC input primarily modulates the valence of a tastant, and that integration of other sources of information about the US may be necessary for eliciting a strong aversion.

Recent work has identified a role for corticoamygdala projecting neurons in CTA (*Lavi et al., 2018*; *Kayyal et al., 2019*), suggesting that this population of GC neurons is involved in the acquisition and retrieval of the aversive taste memory. Whether the population of amygdala-projecting and amygdala-recipient neurons in GC overlap is currently unknown, but this study raises interesting questions about how and when reciprocal connections between BLA and GC are recruited in CTA. It is also unclear whether amygdalocortical neurons in GC project exclusively to BLA, or also interact with other nodes of the CTA circuit.

## Conclusion

LTD was proposed as a mechanism for learning in cerebellar circuits (*Ito et al., 2014*; *Hirano et al., 2016*) and is thought to contribute to the control of alcohol-seeking behavior in the basal ganglia (*Ma et al., 2018*). In both circuits, LTD is induced at glutamatergic synapses onto GABAergic neurons (*Hansel and Linden, 2000*; *Kakegawa et al., 2018*), providing a form of disinhibition to the downstream synaptic targets. Our study focused on plasticity of BLA input onto excitatory neurons in L2/3 of GC, strongly supporting a role for LTD at glutamatergic synapses onto pyramidal neurons in hedonic learning.

To our knowledge, our results are the first to directly assess the effect of CTA on BLA-GC synapses, and provide the first evidence for a direct link between LTD of amygdalocortical transmission onto GC neurons and taste aversion learning. Taste circuits are highly conserved among species, and changes in taste preference can be induced similarly in humans as well as many other organisms, making CTA and the amygdalocortical system ideal models for investigating neural mechanisms of hedonic learning. Our results link a specific form of LTD at the BLA-GC input to a change in taste preference, emphasizing how modulation of synaptic efficacy underlies adaptive shifts in hedonic value capable of influencing perception and behavior.

## Materials and methods

Long Evans rats of both sexes were used for this study. Animals were singly housed in a vivarium on a 12 hr/12 hr light dark cycle with ad libitum access to food and water, except where otherwise noted. Experiments were conducted during the light period. All surgical and experimental procedures were approved by the Institutional Animal Care and Use Committee of Stony Brook University and followed the guidelines of the National Institutes of Health.

### Conditioned taste aversion training

Rats were placed on water restriction with free access to food for a total of 8 days. Rats were habituated to a behavioral chamber where they had 15 min access to a drinking spout with $H_2O$, followed by 1 hr access to $H_2O$ in their home cage four hours later. The volume consumed was recorded daily throughout training (total volume (ml): juveniles, 12.75 ± 0.70; adults, 16.96 ± 1.17), and rats' weight was monitored to ensure that it remained within 85% of initial weight. Four days of habituation training was sufficient to stabilize fluid intake levels (Habituation, *Figure 1B,C*). This was followed by two conditioning trials, with a recovery day in between identical to the habituation days. The recovery day enabled us to confirm that the conditioning procedures did not affect thirst levels. For immunohistochemistry and slice electrophysiology, conditioning consisted of 15 min access to a drinking spout with 0.1M sucrose, followed by an ip injection of 0.15M LiCl (7.5 mL/kg) to induce gastric malaise. Rats in the control group received an ip LiCl injection (0.15M) in the evening on day 4 and 6 of training. The injection was delivered approximately 16 hr before sucrose presentation to ensure lack of association between gastric malaise and sucrose consumption (*Figure 1B*, pseudo CTA). For experiments in vivo, conditioning for rats in the GFP control group (*Figure 6C*) and paired 20 Hz opto group (*Figure 6D*) consisted of 15 min access to a drinking spout with 0.1M sucrose, followed by optogenetic 20 Hz stimulation of BLA terminal fields in GC. Rats in the non-paired 20 Hz opto group (*Figure 6E*) received 20 Hz stimulation of BLA terminal fields in GC the evening prior to sucrose exposure. Rats in the ramp opto group (*Figure 6F*) had sucrose exposure followed by optogenetic ramping stimulation of BLA terminal fields in GC. On the 8th training day, all groups engaged in a 2-bottle test to assess a preference for $H_2O$ or sucrose. In a subset of experiments, the 2-bottle test was repeated on day 9, to assess whether the first test had initiated an extinction process (*Figure 1—figure supplement 2*).

### Immunohistochemistry

Detection of immediate early gene expression - 1 hr after the 2-bottle test, rats were anesthetized and intracardially perfused with PBS followed by 4% paraformaldehyde in PBS (4% PFA). The brain was dissected out and thin (50 μm) coronal slices containing GC were cut with a fixed tissue vibratome (Leica VT1000). Sections were washed in PBS (3 × 10 min rinse), permeabilized and blocked in a solution containing 0.5% Triton X and 10% normal goat serum in PBS for 1 hr, then incubated

overnight at 4°C in a solution containing 0.1% Triton X and 3% normal goat serum in PBS, mouse anti-GAD67 (1:500, MilliporeSigma, MAB5406, monoclonal), and either rabbit anti-c-Fos (1:500, Cell Signaling, 2250S, monoclonal) or rabbit anti-EGR1 (1:500, Cell Signaling, 4153S, monoclonal). Sections were then rinsed in PBS (3 × 10 min) and incubated at 25°C for 2 hr in a solution containing 0.1% Triton X and 3% normal goat serum in PBS, goat anti-mouse Alexa Fluor-647 (1:500, Invitrogen, A21235), goat anti-rabbit Alexa Fluor-568 (1:500, Invitrogen, A11011), and NeuroTrace 435/455 (1:1000, Invitrogen, N21479). After a final wash in PBS (3 × 10 min), sections were mounted with Fluoromount-G (Southern Biotech). Sections were imaged with a confocal microscope (Olympus Fluoview) at 20x magnification. Four sections, spaced at 200 µm, were counted per animal for each IEG (c-Fos = 4, EGR1 = 4) using ImageJ software by a person blind to experimental condition. Cell counts represent the number of IEG-positive, GAD67-negative neurons, normalized by the counting area (L2/3 aGC) for each section.

Recovery of recorded neurons in acute slice preparation - Recorded slices were fixed in 4% PFA for 1 week. They were then washed in PBS (3 × 10 min rinse), permeabilized and blocked in a solution containing 1% Triton X and 10% normal goat serum in PBS for 2 hr, then slices were incubated overnight at 4°C in a solution containing 0.1% Triton X and 3% normal goat serum in PBS, streptavidin Alexa Fluor-568 conjugate (1:2000, Invitrogen, S11226), mouse anti-GAD67 (1:500, MilliporeSigma, MAB5406, monoclonal), and chicken anti-GFP (1:1000, Abcam, ab13970, polyclonal). GC slices were then rinsed in PBS (3 × 10 min) and incubated at 25°C for 2 hr in a solution containing 0.1% Triton X and 3% normal goat serum in PBS, goat anti-mouse Alexa Fluor-647 (1:500, Invitrogen, A21235), goat anti-chicken Alexa Fluor-488 (1:500, Abcam, ab150173), and Hoechst 33342 (1:5000, Invitrogen, H3570). BLA slices were incubated overnight at 4°C in a solution containing 0.1% Triton X and 3% normal goat serum in PBS and chicken anti-GFP (1:1000, Abcam, ab13970, polyclonal). They were then rinsed in PBS (3 × 10 min) and incubated at 25°C for 2 hr in a solution containing 0.1% Triton X and 3% normal goat serum in PBS, goat anti-chicken Alexa Fluor-488 (1:500, Abcam, ab150173), and Hoechst 33342 (1:5000, Invitrogen, H3570). Validation of injection sites and positioning of fiber optics - Following behavioral training, animals were anesthetized and perfused intracardially with PBS followed by 4% PFA. Coronal slices containing BLA and GC were cut at 100 µm with a fixed tissue vibratome (Leica VT1000). All sections were washed in PBS (3 × 10 min rinse) and incubated at 25°C for 1 hr in a solution containing 0.1% Triton X and Hoechst 33342 (1:5000, Invitrogen, H3570). Sections were mounted with Fluoromount-G (Southern Biotech) and the accuracy of viral injections and fiber placements was assessed.

## Electrophysiology

Acute coronal slices containing GC were obtained immediately following the 2-bottle test. Slice preparation was as described in our previous study (Haley et al., 2016). Briefly, rats were anesthetized with isoflurane (Bell jar to effect) and rapidly decapitated. The brain was dissected in ice cold, oxygenated artificial cerebrospinal fluid (ACSF), and 300 µm coronal slices containing GC were prepared using a fresh tissue vibratome (Leica VT1000) starting at 1.5 mm to bregma. Patch clamp recordings were obtained from visually identified L2/3 pyramidal neurons under DIC optics. Their identity was tested online with the injection of square current pulses (700 ms) to assess regular firing patterns, and post-hoc with immunohistochemistry aimed at reconstructing neuron morphology, assessing location, and confirming lack of expression of the GABA neuron marker GAD67. To assess possible changes in intrinsic excitability induced by CTA, recordings were obtained in current clamp and square current pulses of increasing amplitude were injected to examine input resistance, action potential threshold, and frequency/intensity curves. Input resistance was calculated from the linear portion of the voltage response to a −50 pA current injection. Action potential threshold was calculated as the membrane voltage when the first derivative of the voltage trace dV/dt = 20 V/s (Bekkers and Delaney, 2001). In a subset of experiments intrinsic properties were assessed in current clamp in the presence of fast synaptic receptor blockers (in µM: DNQX, 20; APV, 50; picrotoxin, 20). In a different set of experiments, spontaneous synaptic activity was recorded in voltage clamp. For these experiments, a cesium sulfate-based internal solution containing the sodium channel blocker QX314 (2 mM, Tocris Bioscience) was used to stabilize recordings during prolonged depolarization. Spontaneous excitatory synaptic currents were recorded at three different holding potentials around the reversal potential for chloride (to record sEPSCs, in mV: −55,−50, −45). Current vs. voltage functions were used to identify the voltage that better isolated the current of interest, which

were used for analysis of charge and spontaneous events' amplitude and frequency (*Maffei et al., 2004*; *Haley et al., 2016*). Total charge was calculated by integrating 5 s sections of the current trace. Evoked EPSCs/EPSPs from stimulation of BLA terminal fields were obtained in voltage clamp and current clamp using a potassium gluconate solution and holding neurons at −70 mV. The monosynaptic nature of BLA-EPSCs was verified as in our previous study (*Haley et al., 2016*). Event-triggered average of light-evoked BLA-EPSCs and BLA-EPSPs was used to align BLA-EPSC and BLA-EPSP onset and calculate the average amplitude of the light-evoked response. Latencies of BLA-evoked responses were calculated from the onset of the phasic 5 ms light pulse. For plasticity experiments, 2 LED pulses (5 ms, 10 Hz) were used to elicit BLA-EPSCs every 30 s. A brief (10 ms) 1 mV depolarization step was used to monitor series resistance ($R_s$). After a 10 min baseline, either the phasic or tonic induction paradigm was applied in current clamp, after which BLA-EPSCs were recorded for an additional 40 min. Neurons with $R_s$ >20 MΩ or that changed >15% during recording were excluded from analysis.

## Optogenetics

Ex vivo experiments: Rats (P14) were anesthetized intraperitoneally with a mixture containing 70 mg/kg ketamine, 0.7 mg/kg acepromazine, and 3.5 mg/kg xylazine (KXA) and mounted on a stereotaxic apparatus. Animals received an injection of the AAV9.CAG.ChR2-Venus.WPRE.SV40 construct (*Petreanu et al., 2007*; University of Pennsylvania Vector Core) in the BLA using a nanoject pressure injector (Drummond Nanoject II; 100 nL volume containing $5.64^{12}$ particles/mL) as described in our previous study (*Haley et al., 2016*). The stereotaxic coordinates for the injections were 2.1 mm posterior to bregma; 4.7 mm lateral to midline; 7.0 mm below the pia. Rats were allowed to recover from surgery for 2 weeks before CTA training and recordings. AAV9 was chosen because it is primarily transported anterogradely, and because ChR2 expressed via AAV9 does not alter the short-term dynamics of evoked responses (*Jackman et al., 2014*). The lack of backfilled somata in all of our GC preparations further confirms forward direction of transport and indicates that only BLA terminal fields in GC were activated by light pulses. Consistency of ChR2-Venus expression across preparation was as previously reported (*Haley et al., 2016*). Briefly, as shown in *Figure 3B*, the intensity of the Venus signal in GC was quantified across a 500 µm wide region of interest (ROI) spanning the cortical mantle from the pia to L5/6 (included) across 10 preparations. The average and standard deviation of the fluorescent signal were plotted as a function of depth. The expression of the Venus signal was then assessed for all preparations used for electrophysiological recordings and compared to the calibration curve. Only data obtained from preparations with a fluorescence profile within one standard deviation of the average calibration curve were included in the analysis.

BLA-EPSCs were evoked with a 5 ms light pulse delivered through the 40x objective mounted on an upright microscope (Olympus BX51WI). The intensity, frequency, and timing of the blue light stimuli (470 nm) were controlled with an LED driver (ThorLabs). Intensity at the output was verified with a light meter (Thor Labs, Range: 0.2–4.4 mW). Phasic stimulation for plasticity induction was organized in 20 trains of 20–5 ms light pulses at 20 Hz delivered every 250 ms and paired with depolarization of the postsynaptic neuron with a 50 pA current step. Tonic activation of BLA terminal fields was achieved using a ramping light stimulus (which prevented inactivation of ChR2 channels *Lin et al., 2009*). The duration of the ramping stimulus was 6 s from 0 to maximum (4.4 mW) light intensity and was controlled through the LED driver connected to an analog output of the amplifier. For plasticity induction, 10 ramping stimuli spaced 5 s were paired with depolarization of the postsynaptic neuron with a 50 pA current step.

## In vivo experiments

Stimulation of BLA terminal fields in vivo was achieved through optical fibers (Doric lenses, 400 µm) chronically implanted in GC (1.4 mm anterior to bregma; 5 mm lateral to midline; 4.5 mm below the pia). Rats, weighing >250 g were anesthetized intraperitoneally with the KXA cocktail described above and supplemented with 40% of the induction dose as needed to maintain surgical levels of anesthesia. The scalp was exposed and holes were drilled for anchoring screws, optical fibers, and viral injection. The AAV9.CAG.ChR2-Venus.WPRE.SV40 construct (*Petreanu et al., 2007*) was bilaterally injected in the BLA (3.0 mm posterior to bregma; 5 mm lateral to midline; 7.2 mm below the pia) using a nanoject pressure injector (Drummond Nanoject II; 200 nL volume containing $5.64^{12}$

particles/mL) as described above. Optical fibers (Doric Lenses, 400 µm) were coated with DiI for better detection of the optic fiber tract and implanted bilaterally to target GC and were secured in place with dental cement.

For these experiments, recovery from surgery was 4 weeks prior to CTA training. The injection site in BLA and the positioning of the optic fiber in GC were verified histologically for each preparation. Stimulation was controlled via the digital output (phasic protocol) or analog output (ramp protocol) of a multi-patch clamp amplifier (HEKA) and software (Patchmaster) and delivered through the implanted optical fibers coupled to a blue laser (Shanghai Dream, 470 nm). A control group of rats received a BLA injection of the AAV9.hSyn.eGFP.WPRE.bGH construct (University of Pennsylvania Vector Core) to ensure that possible changes in sucrose preference were due to BLA terminal field stimulation and not to unspecific effects of the light stimulation. Light intensity at the tip of the optic fiber was measured with a light meter (6.5 mW, ThorLabs) The induction paradigm in vivo had the same structure as the plasticity induction pattern used in slices (Phasic: Trains of 20 pulses, in this case 10 ms long, at 20 Hz delivered every 250 ms, repeated 100 times; Ramp: 12 ramping stimuli 6 s long, spaced 5 s). Animals in the GFP control group, paired 20 Hz opto group, and ramp opto group received photostimulation immediately after exposure to the 0.1M sucrose solution on conditioning days 1 and 2. The non-paired 20 Hz opto group received photostimulation the evening prior to sucrose exposure.

## Data analysis

Data were acquired with a HEKA four channels amplifier with integrated acquisition board. Data were sampled at 10 kHz. Patchmaster (WaveMetrix) was used as acquisition software. Analysis was performed with custom procedures in Igor (WaveMetrix), Matlab, and Clampfit (Molecular Devices). For two-group comparisons (paired and unpaired), we used estimation statistics (*Ho et al., 2019*) to report the effect size and calculate the 95% confidence interval of the mean difference via bootstrap resampling (5000 resamples). Graphic visualization of the data includes the group mean and 95% confidence interval, swarm plots of individual data values, and histograms of the resampling distribution of the difference in the means. Multiple comparisons were assessed with One-Way ANOVA followed by post-hoc Tukey HSD post-hoc tests. Cumulative distributions were compared using the 2-sample Kolmogorov-Smirnov (K-S) test. Rank order correlation analysis was used to determine correlations between IEG expression and sucrose consumption (*Figure 1*), BLA-EPSC/BLA-EPSP amplitude (*Figure 3E*, input/output function), and plasticity and PPR, CV, or Decay tau (*Figure 4—figure supplement 1*, *Figure 5—figure supplement 1*). For plasticity induction experiments, differences between BLA-EPSC amplitude, PPR, and CV were assessed in the 10 min baseline period and in the 10 min period 30–40 min post-induction. Differences in probability of LTD vs. LTP induction following CTA were assessed with a $3 \times 2$ $\chi^2$ for contingency test. Significant differences were considered for $p<0.05$.

## Solutions

ACSF contained the following (mM): 126 NaCl, 3 KCl, 25 NaHCO3, 1 NaHPO4, 2 MgSO4, 2 CaCl2, 14 dextrose with an osmolarity of 315-325mOsm. The internal solution for analysis of excitability and evoked synaptic responses was as follows (mM): 116 K-Glu, 4 KCl, 10 K-HEPES, 4 Mg-ATP, 0.3 Na-GTP, 10 Na-phosphocreatine, 0.4% biocytin ($V_{rev}$ [$Cl^{-1}$] = −81 mV). The pH of the internal solution was adjusted to 7.35 with KOH and the osmolarity adjusted to 295mOsm with sucrose. The internal solution to assess spontaneous excitatory and inhibitory charge and events contained (mM): 20 KCl, 100 Cs-sulfate, 10 K-HEPES, 4 Mg-ATP, 0.3 Na-GTP, 10 Na-phosphocreatine, 3 QX-314 (Tocris Bioscience), 0.2% biocytin (Vrev [Cl−1] = −49.3 mV). *Drugs*. To assess the monosynaptic nature of BLA-EPSCs recordings were obtained in the presence of (in µM): 1 TTX (Tocris Bioscience), 100 4-aminopyridine (4-AP) (Tocris Bioscience). To determine possible changes in intrinsic excitability, the following blockers were bath applied (in uM): 20 DNQX (Tocris Bioscience), and 50 APV (Tocris Bioscience), 20 picrotoxin (Tocris Bioscience).

## Acknowledgements

This work was supported by NIH-NIDCD awards DC013770, DC015234, NS115779 to AM and AF. We wish to thank Dr. Antonello Bonci for useful discussions and feedback on the study.

## Additional information

### Funding

| Funder | Grant reference number | Author |
|---|---|---|
| NIH Blueprint for Neuroscience Research | DC013770 | Alfredo Fontanini Arianna Maffei |
| NIH Blueprint for Neuroscience Research | DC015234 | Alfredo Fontanini Arianna Maffei |
| NIH Blueprint for Neuroscience Research | NS115779 | Alfredo Fontanini Arianna Maffei |

The funders had no role in study design, data collection and interpretation, or the decision to submit the work for publication.

### Author contributions

Melissa S Haley, Conceptualization, Data curation, Formal analysis, Validation, Investigation, Visualization, Methodology, Writing - original draft, Writing - review and editing; Stephen Bruno, Data curation, Formal analysis, Investigation, Writing - original draft, Writing - review and editing; Alfredo Fontanini, Conceptualization, Funding acquisition, Validation, Investigation, Writing - original draft, Writing - review and editing; Arianna Maffei, Conceptualization, Supervision, Funding acquisition, Validation, Investigation, Visualization, Writing - original draft, Project administration, Writing - review and editing

### Author ORCIDs

Melissa S Haley (iD) https://orcid.org/0000-0002-6018-2809
Alfredo Fontanini (iD) http://orcid.org/0000-0003-4561-9563
Arianna Maffei (iD) https://orcid.org/0000-0001-6755-8904

### Ethics

Animal experimentation: All surgical and experimental procedures were approved by the Institutional Animal Care and Use Committee (IACUC) of Stony Brook University and followed the guidelines of the National Institutes of Health. Experiments were done following procedures described in the IACUC protocol # 208916 currently approved until 11/01/2021.

### Decision letter and Author response

Decision letter https://doi.org/10.7554/eLife.55175.sa1
Author response https://doi.org/10.7554/eLife.55175.sa2

## Additional files

### Supplementary files
• Transparent reporting form

### Data availability

All data generated or analyzed during this study are included in the manuscript. Source files data have been provided for all of the Figures and Figure supplements. Legends for Figures and corresponding Source Data Files have been provided. In the Source Data files we have also added information about the statistical tests used for each data set and the corresponding values.

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
