## [Decision Letter]

**Acceptance summary:**

This paper investigates the cellular basis of conditioned taste aversion, the avoidance of a taste that previously produced malaise. The authors first show that optogenetic stimulation of basolateral amygdala (BLA) neurons produces long-term synaptic depression. They then paired the delivery of the taste with this same optogenetic BLA stimulation, and report that rats develop conditioned taste aversion just as if the taste had been paired with an emesis-inducing stimulus. Thus, for the first time, this paper identifies circuitry and synaptic plasticity that underlies the affective dimension of a sensory stimulus.

**Decision letter after peer review:**

Thank you for submitting your article "LTD at amygdalocortical synapses as a novel mechanism for hedonic learning" for consideration by *eLife*. Your article has been reviewed by three peer reviewers, one of whom is a member of our Board of Reviewing Editors, and the evaluation has been overseen by Laura Colgin as the Senior Editor. The following individual involved in review of your submission has agreed to reveal their identity: Donald B Katz (Reviewer #2).

The reviewers have discussed the reviews with one another and the Reviewing Editor has drafted this decision to help you prepare a revised submission.

Summary:

In this manuscript, the Maffei lab brings a combination of slice recordings, behavior, and immunohistochemistry to bear light on the nature of conditioned taste aversion-related plasticity. They first show that CTA decreases overall activity levels of gustatory cortical (GC) neurons, and then go on to provide several pieces of evidence supporting the suggestion that this decrease may be a function of learning-related LTD at amygdala-cortical synapses. Finally, and in the manuscript's fanfare moment, they show that rats get CTAs when the same optogenetic stimulation of BLA neurons that causes learning-appropriate LTD is paired with taste delivery as a substitute for emesis. The results are intriguing and will be of broad interest to those who study synaptic plasticity, amygdala, and the control of taste preference.

Essential revisions:

1) Throughout the paper, the authors report numbers of slices and numbers of animals, however it appears that the slice number is the variable used at least in Figure 6, where the animal N=2. Since the independent variable in the experiment is at the level of the animal, not the slice (i.e. taste aversion training vs. no aversion training), the appropriate n is the animal number, not the slice number (if several slices/cells were used, then those from a single animal can be averaged to make a single animal N). In slice experiments where the manipulation affects only the slice (e.g., if a drug were added to a slice compared to slices receiving vehicle; in this case, the animals were treated identically), it is appropriate to use slice number as the n. The statistics in the paper should be re-calculated with this consideration in mind, and in some figures such as Figure 6, more experiments will be required.

2) There were concerns about the key experiment in which CTA was caused by a pairing of taste with BLA stimulation. Further experiments would nail down more thoroughly the mechanistic aspects and/or controls.

First, the interpretation of this experiment relies on the assumption that associative plasticity was caused by pairing of the inputs. This interpretation would be greatly strengthened by an additional control group, in which it is shown that learning is not caused by NON-CONTINGENT stimulation of rats infected with the ChR2-Venus virus.

Secondly, the connection between the LTD protocol in vitro and the modification of taste preference would be strengthened if the authors made more of a connection, perhaps showing that a drug that blocks the LTD also prevents the modulation of taste preference. At least it seems important to do other control behavioral tests to provide more evidence that links the plasticity to learning the aversion. Is it possible the animals can't taste sucrose as well or don't find it as palatable when the 20Hz stimuli are used? Or are unable to respond for some other reason? Do other tastants work?

The reviewers felt that one of these experiments would have the potential to change or cement the results.

---

## [Author Response]

Essential revisions:1) Throughout the paper, the authors report numbers of slices and numbers of animals, however it appears that the slice number is the variable used at least in Figure 6, where the animal N=2. Since the independent variable in the experiment is at the level of the animal, not the slice (i.e. taste aversion training vs. no aversion training), the appropriate n is the animal number, not the slice number (if several slices/cells were used, then those from a single animal can be averaged to make a single animal N). In slice experiments where the manipulation affects only the slice (e.g., if a drug were added to a slice compared to slices receiving vehicle; in this case, the animals were treated identically), it is appropriate to use slice number as the n. The statistics in the paper should be re-calculated with this consideration in mind, and in some figures such as Figure 6, more experiments will be required.

We understand the concern. To address this, where appropriate, we have averaged cells from individual animals and reassessed statistical significance using the number of animals. Additional experiments have been performed to increase the number of animals for the data presented in Figure 6. With one exception (BLA-EPSP amplitude, Figure 4D, which now shows a trend) the effects we report remain statistically significant when calculated at the level of the animal instead of the cell. In addition to standard statistical testing, we have also reevaluated our data using estimation statistics (Ho et al., 2019), a method that provides additional information on the effect size and confidence values. We now show estimation plots overlapped to our bar graph and we report confidence intervals.

2) There were concerns about the key experiment in which CTA was caused by a pairing of taste with BLA stimulation. Further experiments would nail down more thoroughly the mechanistic aspects and/or controls.First, the interpretation of this experiment relies on the assumption that associative plasticity was caused by pairing of the inputs. This interpretation would be greatly strengthened by an additional control group, in which it is shown that learning is not caused by NON-CONTINGENT stimulation of rats infected with the ChR2-Venus virus.

We agree with the reviewer that the use of non-contingent stimulation may help strengthen our conclusions. To address this, we added a set of experiments in which the phasic 20Hz pattern of optogenetic stimulation of BLA axons was delivered non-contingent to sucrose consumption. The data are now reported in Figure 6E. We show that non-contingent stimulation does not affect sucrose palatability.

Secondly, the connection between the LTD protocol in vitro and the modification of taste preference would be strengthened if the authors made more of a connection, perhaps showing that a drug that blocks the LTD also prevents the modulation of taste preference. At least it seems important to do other control behavioral tests to provide more evidence that links the plasticity to learning the aversion. Is it possible the animals can't taste sucrose as well or don't find it as palatable when the 20Hz stimuli are used? Or are unable to respond for some other reason? Do other tastants work?The reviewers felt that one of these experiments would have the potential to change or cement the results.

We understand the concern of the reviewers. Blocking LTD in GC using a pharmacological approach is currently not feasible, as the cellular mechanisms for this form of plasticity in this brain region are not well understood and we cannot predict the site of action of a drug infused directly in GC at the moment. However, in our previous study (Haley, Fontanini and Maffei, 2016), we examined the effect of phasic and tonic stimulation of BLA afferents. The manuscript focused on BLA-GC connectivity and on how the circuit in GC is recruited by the phasic and tonic patterns of optogenetic stimulation used to induce plasticity in the current study.

Based on our plasticity induction experiments, we know the ramp stimulus induced a form of LTD that differed from the 20Hz protocol and was not affected by CTA. To address the reviewers’ comment, we have performed an additional set of experiments in which the ramp stimulus was delivered in GC following exposure to sucrose. Association of sucrose with ramp stimulation of BLA terminal fields in GC did not affect sucrose preference in the 2-bottle test. These data are now presented in Figure 6F. These results demonstrate that LTD induced by phasic stimulation of BLA-GC afferents is sufficient to change sucrose palatability, while LTD induced by tonic BLA-GC input does not.